 SciPost Phys. Lect.Notes 35 (2022)

# Magnetisation and mean field theory in the Ising model

**Dalton A. R. Sakthivadivel**

Department of Mathematics, Stony Brook University, Stony Brook, NY, 11794-3651
Department of Physics and Astronomy, Stony Brook University, Stony Brook, NY, 11794-3800

⋆ dalton.sakthivadivel@stonybrook.edu

## Abstract

In this set of notes, a complete, pedagogical tutorial for applying mean field theory to the two-dimensional Ising model is presented. Beginning with the motivation and basis for mean field theory, we formally derive the Bogoliubov inequality and discuss mean field theory itself. We proceed with the use of mean field theory to determine a magnetisation function, and the results of the derivation are interpreted graphically, physically, and mathematically. We give a new interpretation of the self-consistency condition in terms of intersecting surfaces and constrained solution sets. We also include some more general comments on the thermodynamics of the phase transition. We end by evaluating symmetry considerations in magnetisation, and some more subtle features of the Ising model. Together, a self-contained overview of the mean field Ising model is given, with some novel presentation of important results.

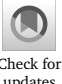
# 1 Introduction

The Ising model is a model of the lattice of particles constituting the atomic structure of a magnetic metal. The model takes a metallic element as being composed of a $d$-dimensional regular lattice $\Lambda$ of atoms, and these atoms as being composed of protons and neutrons. It then models the magnetic properties of certain metals as arising from a *nuclear magnetic moment*, where spin is a quantum mechanical property of particles [1] that creates a magnetic dipole [2,3, especially chapter 3]. Spin $s$ in the Ising model comes in values of either 'up' or 'down,' and each atom (lattice site) has its own individual value of spin, denoted by $+1$ and $-1$, respectively. This integer spin matches that of Bose-Einstein statistics. Details on the nature of spin and its role in the quantum electrodynamical theory of magnets are out of the scope of this article, but roughly speaking, the particle has a charge existing over some volume; this volume 'rotates' on account of the particle's spin, and the moving charge generates a magnetic moment. The alignment of spins creates a strong magnetic moment by amplifying it.

Devised in 1920 by Wilhelm Lenz, Ernst Ising's doctoral supervisor, the model was given to Ising and solved by him in 1925 [4]. Ising's task was presumably to study the properties of the model by solving it, or, to find the model's dynamics from its Hamiltonian. A crucial feature of these dynamics is that the Ising model exhibits a phase transition [5, chapters 5 and 6], wherein the 'phase' of the model changes depending on the heat flow into the model. In this case, the Ising model shows a loss of magnetisation as a temperature variable $T$ increases, corresponding to a real-life phenomenon in which magnets lose their magnetic properties when heated. Ising did eventually solve a one-dimensional model, calculating an expression for the behaviour of a chain of spins from its partition function and determining the spin-spin correlations and free energy [6]. The solution is simple, but unfortunately, there is no phase transition in one dimension, making the classical one-dimensional Ising model uninteresting. In two dimensions, we observe the previously mentioned phase transition from a paramagnet—disordered spins, no magnetisation—to a ferromagnet, at temperatures below a critical point $T_c$. However, in two dimensions, the interactions become too complex to solve for analytically with any ease. In three dimensions, the model is still unsolved.

The Ising model is one of the most commonly used models in statistical mechanics, due both to this phase transition and the richness of the model as a statistical mechanical sandbox; as such, it is important to understand it, despite the challenges it can pose. With respect to its utility in modelling phase transitions, at least, the model has the ability to describe the dynamics of a large number of seemingly quite different transitions. The Ising model thus comprises a particular universality class [7], which describes the grouping of many different systems according to some key common features in their phase transitions [8]. The Ising model has been used for diverse purposes, from describing the liquid-gas critical point to representing various features of string theory [9–12]. This is remarkable for such a simple model. Indeed, the simple appearance, but non-trivial dynamics, of the model make it valuable to statistical mechanics for other reasons than just universality—in particular, the ability to probe difficult statistical phenomena using intuitive equations.

Statistical mechanics is—from one point of view—a science that relates microscopic things to macroscopic things, as in collective phenomena of many-body systems. It does so by simplifying the characterisation of difficult problems with many interacting components, using probabilistic descriptions of the dynamics of these systems. In addition to fundamental notions like ensemble properties, thermal equilibrium, entropy, Boltzmann distributions over states, and so forth, one such statistical method is mean field theory (MFT). MFT constructs a mean field by simplifying the description of a system to an average. Formally, MFT is a way of approximating intractable systems with a simpler model that captures the relevant dynamics of the system. In any stochastic system, each possible state of some variable is observed with some probability,

drawing from a distribution described by a variance around some mean. Likewise, each microstate of the system is observed with a certain probability, such that the configuration of the system admits thermal fluctuations centred around the mean of some distribution describing the microstates. Thus, assuming the so-called relevant dynamics are contained in the spatial average of the system, we can average out rapid fluctuations by assuming they vanish. In so doing, we reduce the degrees of freedom of the system, and are left with a *mean* field. If these fluctuations are caused or amplified by interactions, then this mean field reduces the original theory to an uncoupled field, or a one-body problem. MFT is the primary device of statistical mechanical enquiry in [13], especially chapter five, where the focus is on gaseous bodies of interacting particles. It is closely related to the idea of the renormalisation group, which reduces degrees of freedom by defining a lowermost spatial scale-of-interest, thus constructing an effective field at a particular spatial or energy scale [14]. In both cases, we simplify the problem by restricting ourselves to some notion of a 'relevant' physics.

The simplifications we can make using MFT are inherently an approximation of the true system, and even if useful, they are not always accurate. In fact, renormalisation group methods are a more reliable way to approximate many systems—in the Ising model, for instance, when $d < 4$ or $T = T_c$ [15], the fluctuations we have neglected become too large to actually neglect [16, chapter 12.13]. In the latter case, this leads to a breakdown in the mean field. Mean field theory is known to be inaccurate around such divergences in the order parameter [17]. On the other hand, however, at and above the 'upper critical dimension' $d_c$, a system is so large that the mean field approach gives exact results (e.g. the predicted critical exponents). Notably, these results are only *analytically* exact for $d > d_c$, with logarithmic corrections to the mean field required at $d = d_c$. These are numerically small and thus difficult to detect, but are evident in analysis by renormalisation group. Finite size effects are another problem for the mean field, since MFT implicitly assumes the size of $\Lambda$ is effectively infinite. As such, small corrections are needed when $\Lambda$ is bounded, due to the finite size of the lattice [18]. Whilst for $d \geq d_c$ these finite-size effects are still not trivial [19], mean field results still hold there in general. Along these lines, it has been proven that the dynamics of an Ising model in $d$ dimensions necessarily converge to those of a mean field as $d \to \infty$ [20]. We have already stated that the solution to the Ising model is particularly difficult, and methods to exactly solve it are often lengthy or require ingenuity [21]. With that said—it is clear that, even though MFT is an approximation, it is a principled and useful method, and it holds for many large systems. We will see that, crucially, we can quantify the effects of our approximation by the Bogoliubov inequality.

As outlined, the Ising model exhibits a phase transition, describing the loss of magnetisation that occurs when magnetic materials are heated. Here, we discuss this phase transition, and use MFT as a way of relating the complex microscopic dynamics of spin to the emergent, macroscopic variable of magnetisation. We begin by stating the statistical theory supporting mean field theory, by proving the Bogoliubov inequality. We follow this by focusing on the most formal application of the mean field technique to the Ising model, using the foundation built by the previous formal techniques; later, we will interpret the result that we arrive at.

## 2 Mean Field Theory

### 2.1 Proving the Bogoliubov inequality

MFT is formalised by applying the Bogoliubov inequality to a variational Hamiltonian. Broadly, this states that the choice of a simpler model, and the statistics it yields, can be made formally based on minimisation of a variational term. We explore the Bogoliubov inequality below.

MFT relies on a factorisation of the Hamiltonian into easy and difficult parts. Such perturbative methods are commonly used in physics when a problem is intractable. To treat a system perturbatively, we take a simpler, exactly solvable model, and 'perturb' it with some additional terms to describe the more complicated problem that interests us. In general, these are composed of a solvable expression $A_0$ and an expansion in some control parameter $\lambda$, such that a full equation $A$ is approximated by the series:

$$A \approx A_P = A_0 + \lambda A_1 + \lambda^2 A_2 \dots \lambda^n A_n \,.$$

The Bogoliubov inequality operates on one such perturbative method, and is used to justify MFT. Say we were trying to find the free energy $F$ of a system, given by $-\frac{1}{\beta} \ln\{Z\}$, where this calculation was not tractable due to computational or analytical difficulty. We may separate the system into two components such that one has 'easy' statistics and the other is more complicated, with the caveat that together they must approximate the true Hamiltonian of the system. Free energy is an important statistic, as many fundamental quantities can be derived from it; so, it is natural to ask the question: how good is our approximation of the system's dynamics? The Bogoliubov inequality answers these questions for both statistical and quantum mechanical systems.

Suppose we begin with a simple, unperturbed Hamiltonian $\hat{H}_0$ and perturb it with a more complicated expression $\hat{H}_1$ to get $\hat{H}_P = \hat{H}_0 + \lambda \hat{H}_1$. We want to approximate the behaviour of the true system as best as possible, based only on our choice of the second term. This is a variational problem—we are attempting to identify the minimum difference between the dynamics of our perturbative Hamiltonian $\hat{H}_P$ and those of our true Hamiltonian $\hat{H}$ by varying our perturbative components. In fact, we will see the second term doesn't matter at all, and a judicious choice of *trial* Hamiltonian $\hat{H}_0$ will provide a close match to the actual free energy, which we then improve by minimisation.

In the Bogoliubov inequality, the statistics we reproduce are free energy related. The Bogoliubov inequality ensures the effect an approximation has on the free energy can be given a rigorous upper bound, $F_V$. Given that the full free energy $F$ is a concave function of our control parameter $\lambda$, i.e.,

$$\frac{\mathrm{d}^2 F}{\mathrm{d}\lambda^2} \leq 0 \,,$$

for all $\lambda$, such that $F$ is indeed bounded above, the Bogoliubov inequality gives us the following:

$$F \leq F_0 + \langle \hat{H} - \hat{H}_0 \rangle_0 \,, \tag{1}$$

with the term $F_0 + \langle \hat{H} - \hat{H}_0 \rangle_0$ equalling $F_V$. The task of this section will be to prove and interpret this result.

Deriving the Bogoliubov inequality can be made as simple as observing what happens when we have our perturbative Hamiltonian in the partition function. Say we are able to define this perturbation as an *intentional* collection of simple and difficult terms, such that $\hat{H}_P$ is equal to $\hat{H}$, i.e.,

$$\lambda \hat{H}_1 := \lambda \Delta \hat{H} = \hat{H} - \hat{H}_0 \,.$$

Then, the energy approximation truncates, and the partition function $Z$ is

$$\sum e^{-\beta \hat{H}_k} = \sum e^{-\beta \left( \hat{H}_{0,k} + \lambda \Delta \hat{H}_k \right)} \,, \tag{2}$$

with summation over energy levels, or states $\hat{H}_k$ of $\hat{H}$. Such an energy level is given in quantum mechanics by solving $\hat{H} |\psi\rangle = E_k |\psi\rangle$.

Ideally, the term $\lambda \Delta \hat{H}$ is small, and can be treated as a perturbation in the typical fashion. In this way, it is important to pick the most complicated trial Hamiltonian we can work with.

Such a trial Hamiltonian is often chosen so that higher-order terms vanish, or interactions between the objects in the system factorise [22, chapters 2.11 and 3.4].

We further transform (2) using some basic algebra, with the intention of reducing it to the form of a free energy:

$$
\begin{aligned}
\sum e^{-\beta(\hat{H}_{0,k}+\lambda\Delta\hat{H}_k)} &= Z_0 \sum e^{-\beta\lambda\Delta\hat{H}_k} \cdot Z_0^{-1} e^{-\beta\hat{H}_{0,k}} \\
&= Z_0 \langle e^{-\beta\lambda\Delta\hat{H}} \rangle_0 \, .
\end{aligned}
\tag{3}
$$

The first step in the above set of equations is a simple expansion of the summand, decomposing this exponential term into separate terms for $\hat{H}_0$ and $\lambda\Delta\hat{H}$. Note that $Z_0$ is the partition function for $\hat{H}_0$. The second step, however, uses the mean with respect to the Gibbs distribution to simplify to (3). In particular, we note the mean

$$
\langle e^{-\beta\lambda\Delta\hat{H}} \rangle_0
$$

is the mean of $\exp\{-\beta\lambda\Delta\hat{H}_k\}$ with respect to a Gibbs distribution over states of the trial $\hat{H}$,

$$
\frac{1}{Z_0} e^{-\beta\hat{H}_{0,k}} \, .
$$

In this case, it is clear to see how we arrive at (3).

We now use Jensen's inequality, to try and confidently bound calculations on our partition function by something which is easier to work with. A useful general result on convex functions, Jensen's inequality states that, when a function $f$ of a variable $x$ is convex, e.g., $f'' > 0$, the mean $\langle f(x)\rangle$ is always greater than or equal to $f(\langle x\rangle)$. Here, since $e^{-x}$ is convex, we apply Jensen's inequality to (3) to get

$$
Z_0 \langle e^{-\beta\lambda\Delta\hat{H}} \rangle_0 \geq Z_0 e^{\langle -\beta\lambda\Delta\hat{H}\rangle_0}
$$

and simplify to

$$
\ln\{Z\} \leq \ln\{Z_0\} + \beta\langle\lambda\Delta\hat{H}\rangle_0 \, ,
$$

using that $Z_0 \langle e^{-\beta\lambda\Delta\hat{H}}\rangle_0$ is equal to our partition function $Z$, proven earlier by simplifying (2) into (3). Multiplying by $-k_B T$, this is equivalent to

$$
\begin{aligned}
-k_B T \ln\{Z\} &\leq -k_B T \ln\{Z_0\} + \langle\lambda\Delta\hat{H}\rangle_0 \\
F &\leq F_0 + \langle\hat{H} - \hat{H}_0\rangle_0 \, ,
\end{aligned}
$$

since $F = -k_B T \ln\{Z\}$. As such, we recover the bound given in (1), the Bogoliubov inequality.

The term $F_0 + \langle\hat{H} - \hat{H}_0\rangle_0$ is called our *variational free energy*, denoted by $F_V$, and is the resulting free energy from our perturbative or variational model. This is easier to perform calculations on by construction—$F_0$ is the free energy of our simpler $\hat{H}_0$, and we will see that the perturbative terms are not so difficult to work with. It is, in general, greater than or equal to the actual free energy of the system—we only have equality when there is no perturbative component, and $\hat{H} = \hat{H}_0$. In other words, our partition function with respect to our trial Hamiltonian must be our actual partition function. Clearly, that defeats the purpose of using a perturbative method in the first place. We can, on the other hand, assume that $F_V$ is some curve lying above $F$, and minimise it when the term depends on some parameter $\lambda$, to approximate $F$ as closely as possible. We will demonstrate this in the Ising model now.

## 2.2 Deriving a mean field model by variational methods

The Ising model Hamiltonian $\hat{H}$ is given by the following expression:

$$-J\sum_{i,j} s_i s_j - h\sum_i s_i\,, \tag{4}$$

a measurement of the total energy of the system as contributed by all individual spin-spin interactions. We have each $s$ as an atomic spin, either up or down, and $J$ as a coupling term, along with the influence of a magnetic field $h$ on each spin. The lattice sites in $\Lambda$ are indexed by $i$'s and $j$'s, and so the first term indicates a sum over pairs of interacting spins. Suppose we use the trial Hamiltonian

$$\hat{H}_0 = -\sum_i m_i s_i\,, \tag{5}$$

rather than (4), with states $\hat{H}_{0,k}$ being different combinations of spins across $i \in \{1,\dots,N\}$. This is a system with no interactions, whose spins experience only an effective magnetic field $m_i$, perhaps from neighbouring or coupled spins. In fact, we can further simplify to an isotropic magnetic field $m$—in other words, the same in each direction. The term $F_V$ in our earlier Bogoliubov inequality, (1), becomes

$$F_V = F_0 + \left\langle \left(-J\sum_{i,j} s_i s_j - h\sum_i s_i\right) - \left(-m\sum_i s_i\right) \right\rangle_0\,.$$

We will proceed to simplify the variational free energy so as to calculate its minimum with respect to $m$.

First, we distribute the expectation into the sums inside. This does not use the previously mentioned Jensen's inequality, because expectation is a linear operator and sums are not convex functions. This yields

$$\begin{aligned}
F_V &= F_0 + \left\langle -J\sum_{i,j} s_i s_j - h\sum_i s_i + m\sum_i s_i \right\rangle_0 \\
&= F_0 - J\sum_{i,j} \langle s_i s_j \rangle_0 - h\sum_i \langle s_i \rangle_0 + m\sum_i \langle s_i \rangle_0 \\
&= F_0 - J\sum_{i,j} \langle s_i s_j \rangle_0 + (m-h)\sum_i \langle s_i \rangle_0\,.
\end{aligned} \tag{6}$$

We now take the derivative of (6) with respect to $m$, intending to minimise the variational free energy by setting $\frac{\partial F_V}{\partial m} = 0$. By the reasoning in the end of Section 2.1, this will yield the best possible approximation of the actual magnetisation function. This derivative is

$$\begin{aligned}
\frac{\partial F_V}{\partial m} &= \frac{\partial}{\partial m}\left(F_0 - J\sum_{i,j}\langle s_i s_j\rangle_0 + (m-h)\sum_i \langle s_i\rangle_0\right) \\
&= \frac{\partial F_0}{\partial m} - \frac{\partial}{\partial m}\left(J\sum_{i,j}\langle s_i s_j\rangle_0\right) + \frac{\partial}{\partial m}\left((m-h)\sum_i \langle s_i\rangle_0\right).
\end{aligned}$$

We evaluate these terms separately.

$$
\begin{aligned}
\frac{\partial F_0}{\partial m} &= -\frac{\partial \ln\{Z_0\}}{\partial m} \\
&= -\frac{1}{Z_0}\frac{\partial Z_0}{\partial m} \\
&= -\frac{1}{\left(\sum_k e^{-m\sum_i s_i}\right)}\frac{\partial\left(\sum_k e^{-m\sum_i s_i}\right)}{\partial m} \\
&= -\frac{\sum_k s_i\, e^{-m\sum_i s_i}}{\sum_k e^{-m\sum_i s_i}} \\
&= -\langle s_i\rangle_0 .
\end{aligned}
\tag{7}
$$

$$
\begin{aligned}
-\frac{\partial}{\partial m}\left(J\sum\langle s_i s_j\rangle_0\right) &= -J\sum\frac{\partial\langle s_i s_j\rangle_0}{\partial m} \\
&= -J\sum\left(\frac{\partial\langle s_i\rangle_0}{\partial m}\langle s_j\rangle_0 + \langle s_i\rangle_0\frac{\partial\langle s_j\rangle_0}{\partial m}\right) \\
&= -J\sum\left(\frac{\partial\langle s_i\rangle_0}{\partial m}\langle s_j\rangle_0\right).
\end{aligned}
\tag{8}
$$

$$
\begin{aligned}
\frac{\partial}{\partial m}\left((m-h)\sum\langle s_i\rangle_0\right) &= \sum\langle s_i\rangle_0 + (m-h)\frac{\partial\sum\langle s_i\rangle_0}{\partial m} \\
&= \langle s_i\rangle_0 + (m-h)\frac{\partial\langle s_i\rangle_0}{\partial m}.
\end{aligned}
$$

In the first differentiation we use the trial free energy as defined from the trial partition function, given by the Hamiltonian in (5):

$$
Z_0 = \sum e^{-\beta\hat{H}_{0,k}} .
$$

We use our assumption that the partition function is non-interacting, allowing us to use the *thermal average* of spins in what might otherwise be an intermediate step, identified in (7). We choose to convert the equation to this form, rather than finish the calculation. See the end of this section, especially equations (12) and (13), for more remarks on the thermal average.

In the second differentiation we have used another implication of spins being uncorrelated, namely, that $\langle s_i s_j\rangle = \langle s_i\rangle\langle s_j\rangle$. This allows us to use the 'product rule' in the derivative in (8). We also use an effective field that only influences a single neighbour, rather than both. Thus, one of the derivatives vanishes. This is a reasonable assumption within the mean field regime, given the set-up of our lattice and our non-interacting spins.

In the third and final differentiation we have simply applied the product rule and then reduced the sum over the thermal average to the single thermal average that exists for the system. Our final equation looks like:

$$
\frac{\partial F_V}{\partial m} = -\langle s_i\rangle_0 - J\sum_{i,j}\left(\frac{\partial\langle s_i\rangle_0}{\partial m}\langle s_j\rangle_0\right) + \langle s_i\rangle_0 + (m-h)\frac{\partial\langle s_i\rangle_0}{\partial m} .
\tag{9}
$$

We simplify (9) to

$$
\frac{\partial F_V}{\partial m} = \frac{\partial\langle s_i\rangle_0}{\partial m}\left(-J\sum_{i,j}\langle s_j\rangle_0 + (m-h)\right),
\tag{10}
$$

where the two thermal spin averages cancel and the partial derivative $\frac{\partial \langle s_i \rangle_0}{\partial m}$ factorises. Now, solving (10) for $\frac{\partial F_V}{\partial m} = 0$, we have

$$
\begin{aligned}
0 &= \frac{\partial \langle s_i \rangle_0}{\partial m} \left( -J \sum_{i,j} \langle s_j \rangle_0 + (m - h) \right) \\
&= \left( -J \sum_{i,j} \langle s_j \rangle_0 + (m - h) \right),
\end{aligned}
$$

which yields

$$
m = J \sum_{i,j} \langle s_j \rangle_0 + h. \tag{11}
$$

Finally, we rewrite (11) by using an expression for the thermal average $\langle s_j \rangle_0$. Let $\hat{H}_k$ indicate a possible state of $\hat{H}$, indexed by $i$. In general, the thermal or ensemble average of a statistical variable is a weighted sum over the possible states given by $\hat{H}$,

$$
\langle A \rangle = \frac{\sum A_k e^{-\beta \hat{H}_k}}{\sum e^{-\beta \hat{H}_k}}.
$$

Clearly, one may also define this using a partition function, which we will use now to simplify $\langle s_j \rangle_0$. A state $\hat{H}_{0,k}$ is given by a particular combination of spins

$$
m \sum_i s_i,
$$

such that, for a given (fixed) $k$, a state $\hat{H}_{0,k}$ is the sum of individual spin states $\hat{H}_{0,k}(s_i)$. This holds, more generally, for any non-interacting Hamiltonian; in fact, the same logic can be found in the definition of the thermal average used in (7). Using this fact to rewrite our definition of $Z$, and using that sums in exponents decompose multiplicatively, we have

$$
\begin{aligned}
Z &= \sum_k e^{-\beta \sum_i \hat{H}_k(s_i)} \\
&= \left( \sum_k e^{-\beta \hat{H}_k(s_1)} \right) \left( \sum_k e^{-\beta \hat{H}_k(s_2)} \right) \cdots \left( \sum_k e^{-\beta \hat{H}_k(s_N)} \right) \\
&= \prod_i \left( \sum_k e^{-\beta \hat{H}_k(s_i)} \right). \tag{12}
\end{aligned}
$$

Since individual $\hat{H}_k(s_i)$ are given by $m s_i$, when using (12) on the partition function of $\hat{H}_0$, we consider the two possible spin states: $+1$ and $-1$. Doing so, we have

$$
\begin{aligned}
Z_0 &= \prod_i \left( \sum_k e^{-\beta m s_i} \right) \\
&= \prod_i \left( e^{-\beta m(+1)} + e^{-\beta m(-1)} \right). \tag{13}
\end{aligned}
$$

Any reader familiar with hyperbolic functions will recognise (13) as

$$
\prod_i 2\cosh(\beta m),
$$

and using this in our thermal average, it becomes

$$
\begin{aligned}
\langle s_j \rangle_0 &= \frac{1}{Z_0} \sum_k \hat{H}_{0,k}(s_j) e^{-\beta \sum_j \hat{H}_{0,k}(s_j)} \\
&= \frac{1}{Z_0} \left( \sum_k e^{-\beta \hat{H}_{0,k}(s_1)} \right) \ldots \hat{H}_{0,k}(s_j) \left( \sum_k e^{-\beta \hat{H}_{0,k}(s_j)} \right) \ldots \left( \sum_k e^{-\beta \hat{H}_{0,k}(s_N)} \right) \\
&= \frac{(+1)e^{-\beta m(+1)} + (-1)e^{-\beta m(-1)}}{2\cosh(\beta m)} \\
&= \frac{2\sinh(\beta m)}{2\cosh(\beta m)} \\
&= \tanh(\beta m) .
\end{aligned}
$$

Going from line two to three, we have implicitly used the cancelling of factors from the thermal average with factors from $Z_0$. The only factor that does not cancel is

$$
s_j \left( \sum_k e^{-\beta \hat{H}_k(s_j)} \right) ,
$$

for some $j \in \{1, \ldots, N\}$, by virtue of the additional term for spin; this expression simplifies as we have shown in (13). Using what we have arrived at, we can define the model found in (11) as such:

$$
\begin{aligned}
m &= J \sum_{i,j} \langle s_j \rangle_0 + h \\
&= J \sum_{i,j} \tanh(\beta m) + h .
\end{aligned}
\tag{14}
$$

Since (14) is a sum over neighbouring spins, carried through from our initial (4), it cannot be removed by considering the effective field as we did previously; however, because of this effective field, it can be reduced to multiplication by the number of neighbours $z$. Note that, in various places in this derivation, we observe something similar—a somewhat misleading sum over indices that do not exist in the summand. This is an artefact of the sum more properly being the trace of a matrix of spin entries.

Multiplying by the 'coordination number' $z$, our final mean field model of magnetisation is

$$
m = zJ \tanh(\beta m) + h .
$$

## 3 Magnetisation

### 3.1 Magnetisation in mean field theory

For a graphical analysis, it is typically good to reduce the number of free variables that we would have to plot. Here, we have two variables of import: $m$ and $T$. Since the constant $zJ$ can be absorbed into the argument of the function, $\beta = \frac{1}{k_B T}$, and we can realistically assume zero external field $h$, we have the self-consistent equation

$$
m = \tanh\left( \frac{T_c m}{T} \right) .
\tag{15}
$$

Whilst this clearly suggests that $T_c = \frac{zJ}{k_B}$, this is still a difficult equation to make sense of, since there is no obvious way to isolate $m$. In fact, because hyperbolic equations are transcendental,

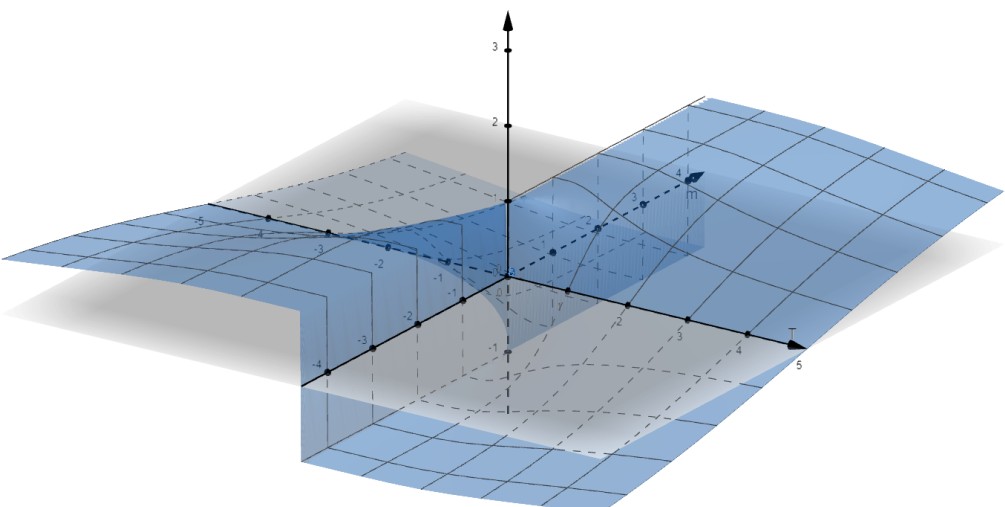

Figure 1: **The magnetisation function of the Ising model.** The surface $f(T, m) = \tanh\left(\frac{T_c m}{T}\right)$ is plotted here for $x = T$ and $y = m$. An interactive version of this figure can be found at https://www.geogebra.org/m/uta3kj9x.

there is *no* way to solve for $m$ algebraically. However, if we rearrange (15) as

$$\tanh\left(\frac{T_c m}{T}\right) - m = 0,$$

we then have the intersection of the magnetisation value with the magnetisation equation in dimensions $T$, $m$, and $f(T, m)$. This is our self-consistency condition: trivially, the intersection between two surfaces is defined by the set of points at which the functions are equal, such that $f(x, y) - g(x, y) = 0$. Thus, we define our self-consistent solution as (15), which also serves as the projection of the intersection of the two surfaces onto the $m$ and $T$ axes. In other words, it projects the intersection of the two surfaces in $f(T, m)$ to $m(T)$, shown in Figure 2. As such, our magnetisation $m(T)$ is this curve, demonstrating the importance of the self-consistency condition.

In order to obtain an expression for magnetisation in terms of temperature, we need to use the previously mentioned self-consistency condition from (15), which evaluates to the intersection of the function $f(T, m) = \tanh\left(\frac{T_c m}{T}\right)$ with $f(T, m) = m$. The self-consistency condition constrains the possible solutions to $m$ as the temperature changes, such that $f(T, m) = m = \tanh\left(\frac{T_c m}{T}\right)$ holds. By applying the self-consistency condition to constrain the evolution of magnetisation in the temperature space, we recover the magnetisation curve for the Ising model.

## 3.2 The thermodynamics of magnetisation

We will explore the physical meaning and intuition for magnetisation in this section. Recall the definition of actual free energy, in statistical mechanical terms:

$$-\frac{1}{\beta} \ln\{Z\}.$$

We want to expand this into thermodynamical terms, so that we can study state variables in a more concise or apparent way, rather than looking at their underlying statistical structure.

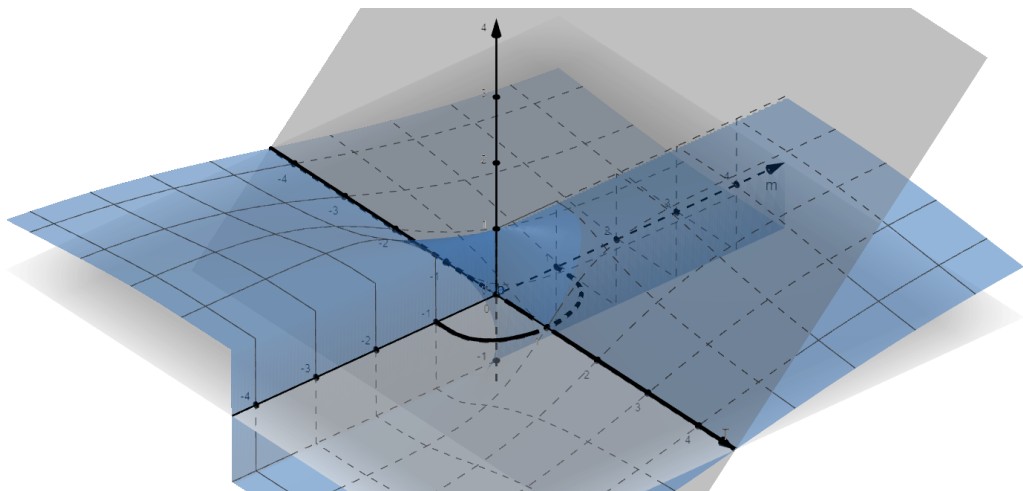

Figure 2: **Plotting magnetisation for temperature using self-consistency.** The surface in grey is $y = m$ and the blue surface is $f(T, m) = \tanh\left(\frac{T_c m}{T}\right)$. The locus of points where $f(T, m) = m$, necessarily given by their intersection, is in the subspace $(T, m(T))$. Indeed, the black curve is the well-known plot of the mean field magnetisation function. An interactive version of this figure can be found at https://www.geogebra.org/m/cjgaepxq.

Beginning with this statistical structure, we use a generalisation of Boltzmann's famous law for ensemble entropy as being proportional to the *number* of microstates $W$,

$$S = k_B \ln\{W\}.$$

In a system coupled to a heat bath we must generalise to the Gibbs entropy, which is the Boltzmann entropy for microstates without equal likelihood. This becomes

$$S = -k_B \sum_i p_i \ln\{p_i\}, \tag{16}$$

which is clearly the Boltzmann equation when $p_i$ is uniform across microstates, e.g., $p_i = W^{-1}$. We condense the probability of energy states into the Boltzmann distribution

$$p_i = \frac{e^{-\beta E_i}}{Z},$$

yielding

$$S = -k_B \sum p_i \left(-\beta E_i - \ln\{Z\}\right)$$

for (16). Distributing the sum over the probability of each state $i$, we get

$$S = k_B \left(\beta \langle E \rangle + \ln\{Z\}\right),$$

from the definition of expectation, $\langle f(x) \rangle = \sum_x f(x) p(x)$. This now gives the following:

$$S = \frac{\langle E \rangle}{T} + k_B \ln\{Z\}$$
$$TS = \langle E \rangle + k_B T \ln\{Z\}$$
$$F = \langle E \rangle - TS.$$

Thus, we have arrived at the macroscopic or Helmholtz free energy. There is an alternate way to do so by using the Laplace transform of an integral over phase space, but that is unnecessary

here. Now, we can analyse what happens to the Ising model, macroscopically, during a phase transition.

Specific to our evaluation of the Ising model, and the simplest consideration of the issue, is that as temperature decreases, entropy contributes increasingly less to the system. This should seem correct, because entropy is reflected in disorder and thermal fluctuations cause disorder. As entropy decreases, our system becomes more ordered. We can look further at this by using the following fact: stable configurations of a system minimise free energy. In that case, we can see by the above that as the entropic contribution to free energy decreases, $F \approx E$. Since the Hamiltonian is a measurement of the total energy of the system, we evaluate this directly, and see that when spins align to $+1$ (in the ferromagnetic case), the system is in the lowest possible energy state. We may calculate this using $\hat{H}$, remembering that the sum occurs over individual multiplicative pairs of spins $i$ and $j$:

$$\hat{H} = -J \sum_{i,j} s_i s_j \, .$$

So, for the lowest possible energy state to occur, every pair of spins must be positive, since this yields a set of positive numbers $s_i s_j$ multiplied by a negative number $-J$. These multiplicative pairs can only be positive when both spins are positive, and hence, every spin must be positive.

This still leaves one question unaddressed: why, in principle, do we expect free energy to be minimised in a stable system? It is as simple as the definition of stability: a stable system does not change, and free energy is the capacity for a system to do work—or, enact change. Thus, when a system does not change, its free energy is minimised. In fact, free energy is *generally* minimised in an equilibrium system, which defines the stable state it can take—even disordered ones. This may seem confusing, but we can also use this to define the tendency to destabilise, and allow entropy to affect spin configurations, in terms of Jaynes' maximum entropy [23]. E T Jaynes defined most results in statistical mechanical dynamics as arising from the natural tendency to maximise entropy, which itself is a simple consequence of the second law of thermodynamics and related phenomena. For high temperatures where $F \approx -TS$, clearly, maximising entropy is equivalent to minimising free energy.

We can define a coherent picture for the energy in this system as follows: the Ising model is attached to a temperature bath, which it is in thermal equilibrium with. We use the following definition of macroscopic entropy: $\Delta S_{\text{bath}} = -T^{-1}Q$. The change in energy towards equilibrium is described by a conservation law, where the heat flow is equivalent to the energy leaving the bath and entering the lattice: $Q = -\Delta E_{\text{bath}} = \Delta E_{\text{IM}}$. As such, the change in entropy towards equilibrium is given by the following:

$$-\frac{Q}{T} + \Delta S_{\text{IM}} \, ,$$

where $Q$ is the heat flow and $\Delta S$ is the change in lattice entropy. We translate this insight from entropy into free energy as follows:

$$\begin{aligned}
\Delta S &= -\frac{Q}{T} + \Delta S_{\text{IM}} \\
&= \frac{-Q + T \Delta S_{\text{IM}}}{T} \\
&= \frac{-\Delta E_{\text{IM}} + T \Delta S_{\text{IM}}}{T} \\
&= \frac{-\Delta (E_{\text{IM}} - T S_{\text{IM}})}{T} \\
&= \frac{-\Delta F}{T} \, .
\end{aligned}$$

Since $\Delta S \geq 0$,

$$\frac{\Delta F}{T} \leq 0.$$

So, regardless of the temperature of the system, an equilibrium state at any $T$ is defined by minimum actual free energy. Once again, at low temperatures, this is accomplished by reducing the actual energy of a configuration, which occurs only when spins are positively aligned. It also allows for stable states at high temperatures, where minimum free energy will come from high entropy.

## 3.3 Broken symmetry and spin configuration

We note one final important consideration for magnetisation and for phase transitions in general. We have previously made reference to a non-zero order parameter as characterising a phase transition. This phenomenological description of the change in macroscopic qualities characterising the phase arises naturally in taking the mean field, but it hides some richer structure than that. In defining what 'order' means, for instance, we are prompted to consider the configuration of the system, and thus symmetry. In a more fundamental sense, an order parameter is a measurement of the breaking of symmetry induced by the phase transition. In zero external field, the Ising Hamiltonian is symmetric under the $\mathbb{Z}_2$ transformation $s \rightarrow -s$, a total rotation:

$$\hat{H} = -\sum J_{ij}(-s_i)(-s_j) \iff \hat{H} = -\sum J_{ij}s_i s_j.$$

This is obviously true, and yet, the ground state of $\hat{H}$ is not symmetric—all spins must align upwards due to the very energy considerations that previously yielded symmetry. To solve this apparent controversy, we re-characterise the phase transition in terms of an order parameter. An order parameter is a measurement of how the physics of a system changes in a phase transition. We can demonstrate that the thermal average of the order parameter vanishes if the symmetries of the Hamiltonian are obeyed. If not, then the order parameter becomes non-zero, and we have broken the symmetry we began with. We saw this directly with magnetisation: $m = 0$ before the phase transition and $m = 1$ afterwards, which is given by the change in configuration from disordered spins to long-range order, the latter of which implies no symmetric mixture of states.

This leads directly to another question, which is more difficult to answer—how is the magnetised state chosen in the non-interacting case, with no symmetry breaking magnetic field $h$? This presents an interesting problem for the mean field approximation, which assumes non-interacting spins, and for a more realistic model where $J$'s are not homogeneous, and the penalty for magnetising 'incorrectly' is not clear. In this case, it is chosen randomly, due to fluctuations at criticality. However, when both states are equally likely, and therefore $\langle m \rangle = 0$, but we expect symmetry to still be broken, a paradox is evident. Indeed, we must ask how we can prove the Ising model will magnetise at all, and what value it is expected to take.

The special case $h = 0$ demands a more sophisticated look at the definition of magnetisation. Degenerate ground states in quantum systems, such as a superposition, or coherence, of possible low energy states, present just this same problem; luckily, there are many methods for describing such systems. The simplest technique for doing so consists of applying the thermodynamic limit $N \rightarrow \infty$ to a perturbative expansion of our degenerate magnetised ground state. For a lattice of size $N$, e.g. the number of sites in $\Lambda$, and positive magnetisation, this method formalises the calculation of the following (non-commuting) double limit:

$$\lim_{h \rightarrow 0^+} \lim_{N \rightarrow \infty} m_N(h). \tag{17}$$

The simplest thing we could do is prove why the limit takes this form. We can do this mathematically by showing that the partition function is a sum of smooth (in fact, analytic) functions

$e^{-\beta \hat{H}_k}$, so for finite $N$, the partition function is also smooth. As such, for $h = 0$, symmetry requires that magnetisation $m(h)$ is continuously zero everywhere. On the other hand, an infinite sum of continuous functions is not necessarily continuous itself, and so admits a discontinuity in $m(h)$ in the case of $h \to 0^+$. In particular, we have $m(h) > 0$ for $h \to 0$ from above. Indeed, a phase transition is defined as a point at which the free energy density becomes non-analytic in the thermodynamic limit, and (17) is the definition of spontaneous magnetisation by symmetry breaking. All of this is summarised by the order parameter becoming non-zero after the critical point.

In addition to proof, it is possible to calculate the perturbative expansion of the ground state of the Hamiltonian. We would see that degenerate ground states couple, or mix, at order $N$, due to $N$ spin-flip corrections in the expansion. Thus, we take $N \to \infty$, so that we have no mixture of ground states in any finite order of the series. The full calculation makes use of quantum field theory and is not quite as trivial as it sounds; in fact, it comes close to solving the Ising model in full. This calculation would eventually take the form of two-point correlation functions, the use of which is one way of finding magnetisation analytically [24, 25]. We can even define spontaneous symmetry breaking in terms of the correlation functions, by showing the existence of off-diagonal long-range order, a hallmark of spontaneous symmetry breaking. We could otherwise define magnetisation as a quasi-average [26], but this also uses some very sophisticated techniques. Clearly, there is more to the story than the mean field approximation—however, these other techniques are very involved. This is a common theme in statistical mechanical systems: knowing anything precisely is often difficult or impossible, and calculations of quantities that are more than approximately known are nearly intractable. This difficulty can even escape the realm of statistical mechanical methods—N N Bogoliubov, responsible for the method of quasi-averages, said, "In modern statistical mechanics, all newly developed methods involve obtaining an understanding and use of the methods of…quantum field theory." This means that principled approximations like mean field theory are incredibly useful for understanding and working with statistical mechanical objects.

## 4 Conclusion

We have derived, from first principles, a mean field theory as a valid approximation for the two-dimensional Ising model. We began by justifying MFT using the Bogoliubov inequality, and then calculated this mean field theory. We then analysed the meaning of these results, and built an intuition for what the mean field magnetisation equation indicates and why the Ising model magnetises at all. We have explored some typical results in a very important statistical mechanical model, and set the stage for even more involved investigation of the Ising model, phase transitions, and statistical mechanics itself.

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
