# Peer review of "Magnetisation and Mean Field Theory in the Ising Model"

_SciPost Physics Lecture Notes, doi:SciPost Phys. Lect. Notes 35 (2022)_

## Round 4 · Referee Report · Anonymous (Referee 1) · 2021-10-16

Report
Mean field theory starts with the basic assumption that the magnetic field acting at any site is the average magnetic moment. With this assumption, one can derive a transcendental equation for magnetic moment (Eq. (14) in the article) following a simple argument, as can be found in any standard text book. The author however arrives at this equation by looking for the value of magnetic moment which minimises the variational free energy. This alternative approach (i) uses “Bogoliubov inequality”, in addition to standard treatments; (ii) is lengthy (iii) adds to our understanding ONLY the knowledge that the conventional mean-field value of magnetic moment minimises the variational free energy. Although I understand that the philosophy of different scientists may be different, I strongly feel that there is no reason why the point (iii) could be helpful to any beginner (student) or researcher (in other areas of physics).
Hence, I stick to my decision of rejection.

---

## Round 4 · Referee Report · Anonymous (Referee 3) · 2021-10-27

Report
by Dalton A R Sakthivadivel. In this paper, the author presented a pedagogical tutorial for the mean-field theory of the two-dimensional Ising model. In addition to discussing the standard mean-field theory, the author makes connections with the Bogoliubov inequality. The results and their implications are discussed both physically and mathematically.
This paper has already been reviewed by two referees twice (the second time being a revised version submitted by the author). One referee had asked for some revisions of the originally submitted version, and eventually recommended acceptance and publication of the revised manuscript, based on the author's response and the revisions. The other referee remained unconvinced about the utility of this work and persisted with his/her recommendation of rejection.
I have carefully read the versions of the paper as well as the referee comments and the corresponding author responses. While it is true that the manuscript largely has known, standard results, I believe it presents the results in a way that might be of some use to students/younger researchers getting initiated into this field.Therefore, on the balance, I recommend publication of the revised manuscript without any change.

---

## Round 4 · Author Response

List of changes
In this resubmission, the following revisions have been made:
Expanded commentary about - Collective phenomena (Introduction) - Fluctuations in MFT and statistical physics more broadly (Introduction) - Order parameters (IIE)
Amendments to - Discussion of renormalisation group and critical dimension (Introduction) - Notation in some areas
All pursuant to reviewer comments. Additionally, a link to an interactive surface plot hosted by the author on GeoGebra has been included, for pedagogical value.

---

## Round 4 · List of Changes

In this resubmission, the following revisions have been made:
Expanded commentary about - Collective phenomena (Introduction) - Fluctuations in MFT and statistical physics more broadly (Introduction) - Order parameters (IIE)
Amendments to - Discussion of renormalisation group and critical dimension (Introduction) - Notation in some areas
All pursuant to reviewer comments. Additionally, a link to an interactive surface plot hosted by the author on GeoGebra has been included, for pedagogical value.

---

## Editorial Decision

published